## Comment

chemical physics/solid-state physics

**Authors for correspondence:**
B. Toudic
e-mail: bertrand.toudic@univ-rennes1.fr
C. Ecolivet
e-mail: claude.ecolivet@univ-rennes1.fr
Mark D. Hollingsworth
e-mail: mdholl@ksu.edu

This article has been edited by the Royal Society of Chemistry, including the commissioning, peer review process and editorial aspects up to the point of acceptance.

# Comment on Couzi *et al.* (2018): a phenomenological model for structural transitions in incommensurate alkane/urea inclusion compounds

B. Toudic[1], L. Guérin[1], C. Mariette[1], I. Frantsuzov[2], P. Rabiller[1], C. Ecolivet[1] and Mark D. Hollingsworth[2]

[1]Univ Rennes, CNRS, IPR (Institut de Physique de Rennes) — UMR 6251, F-35000 Rennes, France
[2]Department of Chemistry, Kansas State University, Manhattan, KS 66506, USA

BT, 0000-0001-9918-8071; LG, 0000-0002-0509-8444; CM, 0000-0001-8067-9591; IF, 0000-0002-3589-6818; PR, 0000-0003-1566-606X; CE, 0000-0002-1929-7655; MDH, 0000-0002-1995-7182

In their recent article [1], M. Couzi *et al.* develop a standard phenomenological model of coupled order parameters, which generates one single symmetry-breaking phase transition. They apply it to the phase transitions of *n*-nonadecane/urea and *n*-hexadecane/urea, while knowing that our measurements had demonstrated, via previously published diffraction experiments [2–6], that both materials undergo at least two symmetry-breaking events revealed by systematic absences of diffraction peaks.

We have shown that in *n*-nonadecane-$d_{40}$/urea-$d_4$ and *n*-nonadecane-$h_{40}$/urea-$h_4$, there is a complex sequence of phases that follow crystallographic symmetry conditions. The first-phase transition at $T_{c1}$ is associated with the symmetry breaking from the hexagonal high-temperature phase to a second phase (phase II); a second transition to a different space group (phase III) occurs at a lower temperature $T_{c2}$. These results have been discussed extensively in the literature [3,4,6–9] and were obtained using excellent spatial resolution and temperature calibration, including measurements using cold neutron scattering on triple axis spectrometers [3,4], and on a synchrotron X-ray diffractometer [7]. For *n*-nonadecane-$h_{40}$/urea-$h_4$, $T_{c1} = (158.8 \pm 0.1)$ K and $T_{c2} = (147.0 \pm 0.1)$ K, according to adiabatic measurements [10].

Instead of using these careful measurements, M. Couzi *et al.* used the results from their recently published article, which describes an X-ray diffraction study performed on $n$-nonadecane-$h_{40}$/urea-$h_4$ but with measurements reported *at only two temperatures* (147 K and 100 K) [11]. In a Comment concerning that article [2], which they did not contest, we demonstrated that their data have no relevance. There [2], we wrote: 'As elaborated here, the data reported by Couzi *et al.* are not from phase II of $n$-nonadecane/urea and cannot be used to discuss the sequence of phases in this compound'.

Inexplicably, Couzi *et al.* do not address this fundamental criticism of their data in the present article [1]. They have not presented any data on pure phase II on which to base their assertion that phase II is the same space group as phase III. Furthermore, M. Couzi *et al.* are fully aware of the extraordinarily high-quality X-ray data exhibiting the unique crystallographic signature of phase II shown in fig. 2(a) of [7], (ref. 32 in [1] and ref. 11 in [11]), which was acquired at 154.5 K, in the middle of phase II of $n$-nonadecane-$h_{40}$/urea-$h_4$. This figure, in fact, is the reconstructed image that they *would* have obtained if they had actually been measuring phase II instead of phase III. However, they have chosen to ignore these data in favour of their own, which were both measurements on phase III.

In the same Comment [2], to which there was no reply, we asked Couzi *et al.* to explain how their measurements were actually made, since *only one* measurement was reported in the vicinity of phase II:

It is for Couzi *et al.* to explain why they have failed to observe the absence/presence conditions that characterize phase II, in particular the common and host superstructure Bragg peaks, which are not in phase II and whose emergence with cooling signifies phase III.

However, in their present article [1], Couzi *et al.* make the following surprising statement (where references [12,28,30] and [31] are references [5,3,4] and [11], respectively, in this Comment):

Given that the $(3 + 2)$-dimensional superspace groups proposed previously [12,28,30] for $n$-nonadecane/urea and $n$-hexadecane/urea have been shown [31] to be incorrect,...

We maintain that the data collected by Couzi *et al.* [11] were on the wrong phase, and as a consequence, their phenomenological description of the phase behaviour in $n$-nonadecane/urea [1] is contrary to reliable experimental measurements and does not apply to the isotopologues of $n$-nonadecane/urea. As already extensively argued in our previous Comment [2], we maintain also that two different symmetry-breaking events are indeed present in $n$-hexadecane/urea.

Data accessibility. This article contains no new data.
Authors' contributions. Each of the authors contributed substantially to the drafting and editing of the manuscript, and all authors approved of the final content.
Competing interests. We declare we have no competing interests.
Funding. No funding has been received for this article.

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
