## [Reviewer comments · Royal Society Open Science]

Review History

RSOS-182073.R0 (Original submission)

Review form: Reviewer 1

Is the manuscript scientifically sound in its present form?

Yes

Are the interpretations and conclusions justified by the results?

Yes

Is the language acceptable?

Yes

Is it clear how to access all supporting data?

Yes

Do you have any ethical concerns with this paper?

Yes

Have you any concerns about statistical analyses in this paper?

No

Recommendation?

Accept with minor revision (please list in comments)

Comments to the Author(s)

This comment submission concerns what appears to be an on-going dispute in the literature over the phase transitions in a family of urea inclusion compounds. These are famous examples of incommensurately modulated structures, and the two "sides" in the dispute include those best known in this area. I refer to the "sides" as Couzi and Toudic below.

As a reviewer it is difficult to appreciate all the complexities of the scientific and non-scientific arguments that have lead to this point. Having read this comment, the papers it refers to and the earlier comment in Europhys Lett [3], my understanding of the situation is that: 1. Couzi et al published reference [2] on "phase III" of an inclusion compound. 2. Toudic et al published a comment in reference [3] presenting evidence that the work had in fact been performed on "phase II". 3. The comment in reference [3] wasn't challenged by Couzi at the time. 4. The RSOS paper by Couzi [1] doesn't properly acknowledge the comment in [3], and still assumes reference [2] was indeed on "phase III", which is at best misleading.

If my understanding in the previous paragraph is correct, then it would seem appropriate that Toudic et al are given an opportunity to publish this comment. I would hope that its content would be shared with Couzi so that both sides get an opportunity to publish their comments and/or rebuttal side-by-side. This would seem the quickest and cleanest way to reach a conclusion and would give minimum confusion for future workers in the field.

I do have some specific comments on the contents of the submission. Some are scientific and some more about the tone of the submission. These latter points are more editorial and I am happy for the editors to decide on what they think is appropriate.

- The comment could perhaps be shorter without losing any content or impact. This would help the reader. I suggest some areas below.
- The crux of the dispute appears to be the temperature at which Couzi's work in reference [2] was performed and the implication on the phase present. The comment contains some reference to phase transition temperatures as TC1/TC2, but it would be helpful to have a short summary of the accepted temperature range for each phase, how it was determined, whether it is disrupted, and whether there is any hysteresis involved. There should then be a clear statement on why they believe Toudic measured phase II. This would allow Couzi et al to explicitly state the reasons why they believe they are in Phase III.
- In paragraph 1 the word "attempt" should be removed. I would suggest changing "arguments rest solely" to "arguments seem to rest solely".
- The final two sentences in paragraph 2 about the "upmost care" and giving details of instruments used by Couzi don't seem necessary at this point. For me they seem to imply the other work wasn't done carefully; it's possible for the other work to be carefully performed even if it proves to be incorrect.
- The wording on P2 "In that Comment...basis for their model" read to me like a letter to the editor rather than a scientific comment. I feel that the key points can be made without directly confrontational language.
- The phrase "simply not true" should be replaced by "incorrect".

To re-emphasise, my comments and recommendation are based on my best understanding of the situation from the information presented, but this is unlikely to be perfect. If this review is shared with either set of authors I would hope they read it in that light.

Review form: Reviewer 2

Is the manuscript scientifically sound in its present form?

Yes

Are the interpretations and conclusions justified by the results?

Yes

Is the language acceptable?

Yes

Is it clear how to access all supporting data?

Not Applicable

Do you have any ethical concerns with this paper?

No

Have you any concerns about statistical analyses in this paper?

No

Recommendation?

Accept as is

Comments to the Author(s)

It is good to show different aspects.

Review form: Reviewer 3

Is the manuscript scientifically sound in its present form?

Yes

Are the interpretations and conclusions justified by the results?

Yes

Is the language acceptable?

Yes

Is it clear how to access all supporting data?

Not Applicable

Do you have any ethical concerns with this paper?

No

Have you any concerns about statistical analyses in this paper?

No

Recommendation?

Accept as is

Comments to the Author(s)

Reasonable argument. I am looking forward to seeing reply from Dr. Couzi et al.

Decision letter (RSOS-182073.R0)

19-Feb-2019

Dear Dr Toudic:

Title: Comment on Couzi et al. (2018): A phenomenological model for structural transitions in incommensurate alkane/urea inclusion compounds

Manuscript ID: RSOS-182073

Thank you for submitting the above manuscript to Royal Society Open Science. On behalf of the Editors and the Royal Society of Chemistry, I am pleased to inform you that your manuscript will be accepted for publication in Royal Society Open Science subject to minor revision in accordance with the referee suggestions. Please find the reviewers' comments at the end of this email. I apologise that this took longer than expected.

The reviewers and handling editors have recommended publication, but also suggest some minor revisions to your manuscript. Therefore, I invite you to respond to the comments and revise your manuscript.

Please also include the following statements alongside the other end statements. As we cannot publish your manuscript without these end statements included, if you feel that a given heading is not relevant to your paper, please nevertheless include the heading and explicitly state that it is not relevant to your work. We have included a screenshot example of the end statements for reference.

- Ethics statement

Please clarify whether you received ethical approval from a local ethics committee to carry out your study. If so please include details of this, including the name of the committee that gave consent in a Research Ethics section after your main text. Please also clarify whether you received informed consent for the participants to participate in the study and state this in your Research Ethics section.

OR

Please clarify whether you obtained the necessary licences and approvals from your institutional animal ethics committee before conducting your research. Please provide details of these licences and approvals in an Animal Ethics section after your main text.

OR

Please clarify whether you obtained the appropriate permissions and licences to conduct the fieldwork detailed in your study. Please provide details of these in your methods section.

Because the schedule for publication is very tight, it is a condition of publication that you submit the revised version of your manuscript before 28-Feb-2019. Please note that the revision deadline will expire at 00.00am on this date. If you do not think you will be able to meet this date please let me know immediately.

Best wishes,
Dr Laura Smith
Publishing Editor, Journals

Royal Society of Chemistry
Thomas Graham House

Science Park, Milton Road
Cambridge, CB4 0WF
Royal Society Open Science - Chemistry Editorial Office

RSC Associate Editor:
Comments to the Author:
(There are no comments.)

RSC Subject Editor:
Comments to the Author:
(There are no comments.)

Reviewer comments to Author:
Reviewer: 1

Comments to the Author(s)

This comment submission concerns what appears to be an on-going dispute in the literature over the phase transitions in a family of urea inclusion compounds. These are famous examples of incommensurately modulated structures, and the two “sides” in the dispute include those best known in this area. I refer to the “sides” as Couzi and Toudic below.

As a reviewer it is difficult to appreciate all the complexities of the scientific and non-scientific arguments that have lead to this point. Having read this comment, the papers it refers to and the earlier comment in Europhys Lett [3], my understanding of the situation is that: 1. Couzi et al published reference [2] on “phase III” of an inclusion compound. 2. Toudic et al published a comment in reference [3] presenting evidence that the work had in fact been performed on “phase II”. 3. The comment in reference [3] wasn’t challenged by Couzi at the time. 4. The RSOS paper by Couzi [1] doesn’t properly acknowledge the comment in [3], and still assumes reference [2] was indeed on “phase III”, which is at best misleading.

If my understanding in the previous paragraph is correct, then it would seem appropriate that Toudic et al are given an opportunity to publish this comment. I would hope that its content would be shared with Couzi so that both sides get an opportunity to publish their comments and/or rebuttal side-by-side. This would seem the quickest and cleanest way to reach a conclusion and would give minimum confusion for future workers in the field.

I do have some specific comments on the contents of the submission. Some are scientific and some more about the tone of the submission. These latter points are more editorial and I am happy for the editors to decide on what they think is appropriate.

- The comment could perhaps be shorter without losing any content or impact. This would help the reader. I suggest some areas below.
- The crux of the dispute appears to be the temperature at which Couzi’s work in reference [2] was performed and the implication on the phase present. The comment contains some reference to phase transition temperatures as TC1/TC2, but it would be helpful to have a short summary of the accepted temperature range for each phase, how it was determined, whether it is disputed,

and whether there is any hysteresis involved. There should then be a clear statement on why they believe Toudic measured phase II. This would allow Couzi et al to explicitly state the reasons why they believe they are in Phase III.

- In paragraph 1 the word “attempt” should be removed. I would suggest changing “arguments rest solely” to “arguments seem to rest solely”.

- The final two sentences in paragraph 2 about the “upmost care” and giving details of instruments used by Couzi don’t seem necessary at this point. For me they seem to imply the other work wasn’t done carefully; it’s possible for the other work to be carefully performed even if it proves to be incorrect.

- The wording on P2 “In that Comment....basis for their model” read to me like a letter to the editor rather than a scientific comment. I feel that the key points can be made without directly confrontational language.

- The phrase “simply not true” should be replaced by “incorrect”.

To re-emphasise, my comments and recommendation are based on my best understanding of the situation from the information presented, but this is unlikely to be perfect. If this review is shared with either set of authors I would hope they read it in that light.

Reviewer: 2

Comments to the Author(s)

It is good to show different aspects.

Reviewer: 3

Comments to the Author(s)

Reasonable argument. I am looking forward to seeing reply from Dr. Couzi et al.

Author's Response to Decision Letter for (RSOS-182073.R0)

See Appendix A.

RSOS-182073.R1 (Revision)

Review form: Reviewer 1

Is the manuscript scientifically sound in its present form?

Yes

Are the interpretations and conclusions justified by the results?

Yes

Is the language acceptable?

Yes

Is it clear how to access all supporting data?

Not Applicable

Do you have any ethical concerns with this paper?

Yes

Have you any concerns about statistical analyses in this paper?

No

Recommendation?

Accept as is

Comments to the Author(s)

I have read the authors comments on, and responses to, my first review. I am happy with all the changes that they have made and strongly recommend prompt publication of this comment.

As I stated in my original review, I hope the comment is shared with Couzi et al so that they have an opportunity to respond to it. Ideally any response would be published alongside the comment. My personal view is that it is important that this comment gets published promptly as the situation seems to have been on-going for some time (I have reviewed within 12 hours for this reason). If the journal goes down this simultaneous-response route I would therefore strongly recommend a short deadline be placed on Couzi et al for response (e.g. 4 weeks including [expedited] review time).

I have to say that I share the authors' feelings about the one-line reviews given by reviewers 2 and 3. Perhaps they made more detailed comments in confidence to the editors? If not, their reviews (particularly referee 2) seem to significantly underestimate the issues being presented in the articles and comments. I was personally disappointed at the apparent lack of critical assessment in these reviews.

Decision letter (RSOS-182073.R1)

01-Mar-2019

Dear Dr Toudic:

Title: Comment on Couzi et al. (2018): A phenomenological model for structural transitions in incommensurate alkane/urea inclusion compounds

Manuscript ID: RSOS-182073.R1

It is a pleasure to accept your manuscript in its current form for publication in Royal Society Open Science. The chemistry content of Royal Society Open Science is published in collaboration with the Royal Society of Chemistry.

The comments of the reviewer who reviewed your manuscript are included at the end of this email. According to the Royal Society's policy on Comments and Replies (<https://royalsociety.org/journals/ethics-policies/editorial-standards/>), the authors of the original article will receive a copy and will be invited to reply. If you have any questions about

this, please contact the Royal Society (openscience@royalsociety.org) and they will be happy to help.

RSC Associate Editor:
Comments to the Author:
(There are no comments.)

RSC Subject Editor:
Comments to the Author:
(There are no comments.)

Reviewer(s)' Comments to Author:
Reviewer: 1

Comments to the Author(s)

I have read the authors comments on, and responses to, my first review. I am happy with all the changes that they have made and strongly recommend prompt publication of this comment.

As I stated in my original review, I hope the comment is shared with Couzi et al so that they have an opportunity to respond to it. Ideally any response would be published alongside the comment. My personal view is that it is important that this comment gets published promptly as the situation seems to have been on-going for some time (I have reviewed within 12 hours for this reason). If the journal goes down this simultaneous-response route I would therefore strongly recommend a short deadline be placed on Couzi et al for response (e.g. 4 weeks including [expedited] review time).

I have to say that I share the authors' feelings about the one-line reviews given by reviewers 2 and 3. Perhaps they made more detailed comments in confidence to the editors? If not, their reviews (particularly referee 2) seem to significantly underestimate the issues being presented in the articles and comments. I was personally disappointed at the apparent lack of critical assessment in these reviews.

Appendix A

Prof. Bertrand Toudic

Institut de Physique de Rennes

F-35042 Rennes

February 25th, 2019

Dear Editor,

We are facing an original and quite distressing experience. For more than 20 years, we have accumulated the highest quality results on the structural instabilities of *n*-alkane/urea inclusion compounds, considering particularly the fully deuterated *n*-nonadecane/urea (deuterated because of the extensive use of coherent neutron scattering). M. Couzi *et al.* have made a single, extremely poor diffraction study on a fully hydrogenated *n*-nonadecane/urea, with only two temperatures measured to characterize three phases! And it appears that we are treated almost equally in terms of the work done. Much more serious, Couzi *et al.* use their measurements at two temperatures to claim in your journal that they have demonstrated that all of our work is incorrect. We maintain that our primary demand is the withdrawal of their article if they cannot demonstrate, as we have asked from the beginning, that their data were measured following a standard scientific protocol that they must provide to the journal.

It is very hard for us to understand why two of the referees limited their responses to one line. Of course, we cannot comment on these reports, simply stating again that we expect that M. Couzi *et al.* will answer the following single question. What control experiments did they perform to ensure that they were indeed measuring phase II of *n*-nonadecane/urea in their study, rather than phase III?

We thank referee 1 for examining our paper carefully. They stress the difficulty of understanding the problem, and write that a shorter comment would be stronger. As a consequence, we have decided to slightly shorten this Comment, focusing on what is most important.

Before responding to the referee's specific comments, we would first like to clarify any confusion there may be about phase II (which exists between T_{c1} and T_{c2}), and phase III (which exists at T_{c2} and below):

a) Couzi *et al.* argue that their measurement at 147 K in the fully hydrogenated *n*-nonadecane/urea is in phase II. They made a second measurement at 100 K and they found that these two measurements were identical from the crystallographic point of view. So they have concluded that phase II and phase III are the same phase.

b) Since Couzi *et al.* reported diffraction patterns at only two temperatures, it appears that they did not try to determine T_{c1}^H (H for the fully hydrogenated *n*-nonadecane-h₄₀/urea-h₄), or at least to calibrate their study since T_{c1}^H is rather well established in the literature ($T_{c1}^H = 158.8$ K). (Note that $T_{c1}^D = 149.4$ K for *n*-nonadecane-d₄₀/urea-d₄).

And more importantly, they have not tried to determine T_{c2}^H , which has never been determined by *anyone* by diffraction. Concerning T_{c2}^H , we must therefore refer to adiabatic calorimetry studies, in which a second transition was reported at $T_{c2}^H = 147.0\text{K}$.

These values are now given in the Comment:

"We have shown that in n-nonadecane-d₄₀/urea-d₄ and n-nonadecane-h₄₀/urea-h₄, there is a complex sequence of phases that follow crystallographic symmetry conditions. The first phase transition at T_{c1} is associated with the symmetry breaking from the hexagonal high temperature phase to a second phase (phase II); a second transition to a different space group (phase III) occurs at a lower temperature T_{c2} For n-nonadecane-h₄₀/urea-h₄, $T_{c1} = (158.8 \pm 0.1)$ K and $T_{c2} = (147.0 \pm 0.1)$ K, according to adiabatic measurements [10]."

*"10. López-Echarri A, Ruiz-Larrea I, Fraile-Rodríguez A, Díaz-Hernández J, Breczewski T, Bocanegra EH 2007. Phase transitions in the urea/n-nonadecane system by calorimetric techniques, J. Phys. Condens. Matter, **19**, 186221. (doi:10.1088/0953-8984/19/18/186221)"*

- In paragraph 1...

We have rewritten the introduction since it appears that the problem was a bit difficult to understand by nonspecialists:

"In their recent article [1], M. Couzi et al. develop a standard phenomenological model of coupled order parameters, which generates one single symmetry-breaking phase transition. They apply it to the phase transitions of n-nonadecane/urea and n-hexadecane/urea, while knowing that our measurements have demonstrated, via previously published diffraction experiments [2-6], that both undergo at least two symmetry breaking events revealed by systematic absences of diffraction peaks."

- The final two sentences in paragraph 2...

In fact the "upmost care" and the details of instruments referred to the work by Toudic *et al.*, not that of Couzi *et al.* We have shortened these lines:

"These results have been discussed extensively in the literature [3,4,6-9] and were obtained using excellent spatial resolution and temperature calibration, including measurements using cold neutron scattering on triple axis spectrometers [3,4], and on a synchrotron X-ray diffractometer [7]."

- The wording on P2...

We agree that the text is written in such a way that it makes reference at the same time to the first Comment in EPL (ref. 2) and to their Article in RSOS (ref. 1). Once again, the difficulty comes from the fact that Couzi *et al.* totally ignored the content of our Comment in EPL and even worse, they used our Comment to say in RSOS that they had demonstrated that we were wrong. (In fact, for us this remains a deliberate lie and an error that the referees of RSOS should have seen.)

To refocus on the present Comment, we have made some slight changes in the new version:

"In the same Comment [2], to which there was no reply, we asked Couzi et al. to explain how their measurements were actually made,..."

"However in their present article [1], Couzi et al. make the following surprising statement..."

- The phrase "simply not true"...

The way Couzi *et al.* treats *n*-hexadecane/urea is very insidious since they did not even try to make any new measurements to verify their interpretation. They simply reinterpret our data in an attempt to demonstrate again that phase II and phase III would be the same, claiming that their Article in EPL constitutes proof, without them introducing any new evidence. We have already discussed this extensively in our Comment in EPL, showing that the sequence of phases in *n*-nonadecane/urea and *n*-hexadecane/urea are totally different. Since we have already answered that point in our previous Comment, and since nothing new concerning *n*-hexadecane/urea has been discussed by Couzi *et al.* in RSOS, we have withdrawn this final part, following the request of the referee to focus on the main points. We, of course, invite the readers to see our previous detailed answer in the Comment in EPL:

"As already extensively argued in our previous Comment [2], we maintain also that two different symmetry breaking events are indeed present in n-hexadecane/urea."

We hope that these modifications, following the referee's advice, will allow the publication of this Comment.

Best Regards,

The Authors